# Identifying effective interventions to promote consumption of protein-rich foods from lower ecological footprint sources: A systematic literature review

Rimante Ronto[1]*, Golsa Saberi[1], Gianna Maxi Leila Robbers[2], Stephanie Godrich[3], Mark Lawrence[4], Shawn Somerset[5], Jessica Fanzo[6], Josephine Y. Chau[1]

**1** Department of Health Sciences, Macquarie University, Sydney, Australia, **2** Nossal Institute of Global Health, University of Melbourne, Melbourne, Australia, **3** School of Medicine and Health Sciences, Edith Cowan University, Perth, Australia, **4** Institute for Physical Activity and Nutrition, Deakin University, Melbourne, Australia, **5** Faculty of Health, University of Canberra, Canberra, Australia, **6** Department of International Health, Johns Hopkins Bloomberg School of Public Health, Baltimore, Maryland, United States of America

* rimante.ronto@mq.edu.au

**Data Availability Statement:** All relevant data are within the paper and its Supporting Information files.

## Abstract

Addressing overconsumption of protein-rich foods from high ecological footprint sources can have positive impacts on health such as reduction of non-communicable disease risk and protecting the natural environment. With the increased attention towards development of ecologically sustainable diets, this systematic review aimed to critically review literature on effectiveness of those interventions aiming to promote protein-rich foods from lower ecological footprint sources. Five electronic databases (Medline, Web of Science, Scopus, Embase and Global Health) were searched for articles published up to January 2021. Quantitative studies were eligible for inclusion if they reported on actual or intended consumption of protein-rich animal-derived and/or plant-based foods; purchase, or selection of meat/plant-based diet in real or virtual environments. We assessed 140 full-text articles for eligibility of which 51 were included in this review. The results were narratively synthesised. Included studies were categorised into individual level behaviour change interventions (n = 33) which included education, counselling and self-monitoring, and micro-environmental/structural behaviour change interventions (n = 18) which included menu manipulation, choice architecture and multicomponent approaches. Half of individual level interventions (52%) aimed to reduce red/processed meat intake among people with current/past chronic conditions which reduced meat intake in the short term. The majority of micro-environmental studies focused on increasing plant-based diet in dining facilities, leading to positive dietary changes. These findings point to a clear gap in the current evidence base for interventions that promote plant-based diet in the general population.

**Funding:** RR funded by Macquarie University New Staff Grant No. 109740183 (https://www.mq.edu.au/). The Funder had no role in study design, data collection and analysis, decision to publish, or preparation of the manuscript.

**Competing interests:** The authors have declared that no competing interests exist.

## Introduction

Current global food systems are not environmentally sustainable [1,2]. Food systems account for 21–37% of anthropogenic greenhouse gas (GHG) emissions and agriculture production for 70% of global freshwater withdrawals [3–5]. Dietary behaviours are both the result and driver of food systems [1]. Unhealthy dietary behaviours have significant impacts on human health, environmental sustainability and contribute to climate change [4]. In order to achieve positive outcomes for human health and the environment, diets that are both healthy and environmentally sustainable are needed.

In 2019, the Food and Agriculture Organization (FAO) of the United Nations and World Health Organization defined sustainable diets as *"healthy dietary patterns that aim to promote optimal health and wellbeing and have minimal environmental pressure and impact. Sustainable healthy diets are equitable, affordable, accessible and culturally acceptable"* [6]. The EAT-Lancet Commission stated that a "planetary healthy diet" consists largely of plant-based foods, low amounts of animal-derived foods (red and processed meat in particular) and little to no added sugars, refined grains, and ultra-processed foods [7]. Addressing overconsumption of animal-derived foods such as red and processed meat can have positive impact on health and environment. Cultivation of animal-derived foods, in aggregate, has a larger environmental impact compared with plant-based food alternatives [8]. Animal-derived foods require more water, more fossil fuels and generate substantially more greenhouse gasses than plant-based food equivalents [9]. Additionally, overconsumption of red meat has been linked to negative health outcomes such as cardiovascular diseases and colorectal cancer [10,11] whereas adequate consumption of fruit and vegetables have protective effects [12]. Processed animal-derived foods have been linked to growing rates of obesity [13] and place burden on natural resources [8].

Growing evidence demonstrates that population level dietary changes can improve health and environmental sustainability and also help in achieving the United Nation's Sustainable Development Goals (SDGs) [14–19]. Changing dietary preferences and behaviours and food systems from animal-derived foods to plant-based diets will require a *'Great Food Transformation'* [7]. There has been an increase in advocacy for and feasibility of harnessing the increasing interest in plant-based diets to influence large population-based public health nutrition interventions, for example Meatless Monday campaigns [20]. However, there is limited evidence on the effectiveness of plant-based dietary behaviour interventions in changing people's behaviour. Therefore, this study aimed to critically review literature on interventions aiming to promote protein rich food intake from low ecological footprint sources to inform the design of larger population-based dietary interventions to achieve major shifts away from a reliance on animal-based foods.

## Materials and methods

The systematic literature review was planned and conducted following the Preferred Reporting Items for Systematic Reviews and Meta-Analyses reporting (PRISMA) statement [21] and the protocol was registered on PROSPERO (Registration Number CRD42020178683). Details of each eligibility criteria are presented in Table 1. This systematic review included quantitative studies only published in peer review academic literature in English language, but it had no restrictions on the study design or year of publication (up until January 2021).

### Search strategy

We searched the following databases: Medline, Web of Science, Scopus, Embase and Global Health. Initially, five primary concepts (meat, plant, food, intake and intervention) were adopted, in order to identify search terms (listed in Table 2). Subsequently, search strings were

**Table 1. Eligibility criteria.**

|  | Inclusion | Exclusion |
|---|---|---|
| **Population** | All except those ones listed in the exclusion criteria. | People diagnosed with clinical condition(s) for which it is required to consume specific amounts of red meat. |
| **Intervention** | Interventions aiming to reduce the demand for red/processed meat and to increase in plant-based proteins including micro-environmental structural (physical) changes. | Dietary interventions aiming to promote a general dietary pattern. Interventions with structural (physical) environment changes but with no evaluation. |
| **Comparator** | No- or minimal-intervention controls, pre-intervention baseline, or other eligible intervention(s). | - |
| **Outcome** | Objective or self-reported measures of demand for red/processed meat and/or plant-based protein, defined as actual or intended consumption, purchase, or selection of meat in real or virtual environments. | - |

developed by the research team and with the help of research librarian. Two researchers (GS and GR) conducted the search independently through all databases. Then, potential articles were imported into Covidence (covidence.org) where duplicates were removed. The screening of search results was conducted and recorded using the PRISMA checklist, by two researchers (GS and GR) independently and in consultation with a third researcher (RR). First, two researchers (GS and GR) independently performed the title and abstract screening of all imported studies against inclusion and exclusion criteria. Where a consensus regarding the inclusion of a study between the first and a second researcher was not reached, it was resolved with two other researchers (RR and SG). Then, full-text versions were obtained for all studies identified to be suitable in the first stage of data screening and reviewed by two researchers (GS and GR) independently. The reference lists of all included studies were hand searched for relevant studies not identified in the first search strategy. Authors of identified studies and experts of the field were consulted, where further details were required. The PRISMA flow diagram was used to document the number of articles at each screening stage (see Fig 1).

**Table 2. Search terms and strings used in a systematic literature review.**

| Number | Search terms and strings |
|---|---|
| 1 | meat OR meat products OR meat proteins OR poultry OR beef OR red meat OR pork OR seafood |
| 2 | (meat OR meatless OR animal derived OR animal OR poultry OR pork OR beef OR seafood).ti,ab. |
| 3 | OR/1-2 |
| 4 | Exp plants/ OR crops, agricultural/ OR plants,.mp OR nuts.mp. OR legumes.mp. OR edible/ OR edible grain/ OR vegetables OR vegetable based.mp. OR vegetar*.mp. OR vegan*.mp. |
| 5 | (plant based OR edible plant* OR vegetable** OR grain).ti,ab. |
| 6 | OR/4-5 |
| 7 | AND/3,6 |
| 8 | Exp Diet/ OR exp Food/ OR exp proteins/ |
| 9 | (food OR diet OR diets OR protein* OR sustainab*).ti,ab. |
| 10 | OR/8-9 |
| 11 | (reduc* OR intake* OR substitut* OR consumption OR "dietary change").mp. |
| 12 | OR/10-11 |
| 13 | AND/7,12 |
| 14 | ((randomized controlled trial OR controlled clinical trial).pt OR randomly.ab. OR trial.ab.) NOT (exp animal/ NOT humans.sh.) |
| 15 | (intervention OR program*).ti,ab. NOT (exp animals/ NOThumans.sh.) |
| 16 | OR/14-15 |
| 17 | AND/13,16 |
| 18 | Limit 17 to English language |

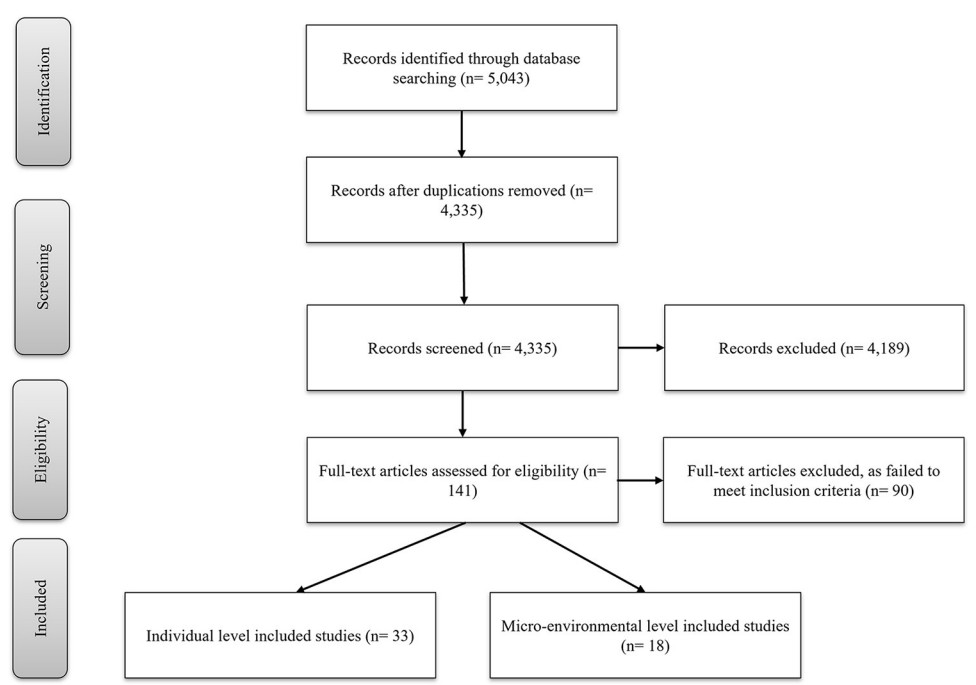

**Fig 1. Flowchart of the literature search and review process.**

## Data extraction and analysis

The following information was extracted from included articles: author(s), year of publication, country of study, title, location, study type (structural/individual), intervention year, intervention length, target audience, sample characteristics, aims, intervention design, behaviour change theory/framework used, eligibility, recruitment, demographic characteristics, measure/tool(s) used, outcomes measured, results, follow up period, follow up results. Three researchers (GS, GR and RR) tested the data extraction sheet by extracting data from 10% of articles; minor disagreements were identified and discussed. Then, two researchers (GS and GR) extracted data from all included studies independently and then cross-checked all extracted data. Any disagreements were resolved in consultation with the third researcher (RR).

The risk and sources of potential bias of each included study were assessed by two researchers (GS and GR) independently by using the Effective Public Health Practice Project Quality Assessment Tool (EPHPP) [22]. This tool was developed to assess the quality of a diverse group of empirical studies. Each included study was assessed on study design, selection bias, confounders, blinding, data collection method, withdrawals and dropouts and assigned to either 'strong', 'moderate' or 'weak' category. Finally, the overall rating was determined based on these ratings as indicated in the assessment tool dictionary. Any disagreements were resolved with a third researcher (RR). Finally, all quantitative data were summarised.

## Results

### Search results and characteristics of included studies

The initial search strategy yielded 5043 studies from five databases. Of these studies, 141 were assessed for eligibility for full-text reviewing of which 90 were excluded due to not meeting the

inclusion criteria. The remaining 51 studies formed the final sample for this review (see Fig 1). These studies were divided into two categories: individual level [23–55] and micro-environmental/structural level [56–73] studies. The summary of each study is provided in Tables 3 and 4.

Of the 33 individual level studies, there were 24 Randomised Controlled Trials (RCT) [23–46], two Non-randomised Controlled Trials (CT) [47,48] and seven used a pre-post study design [49–54]. The number of participants ranged from 7 to 48,835 and the participants' age ranged from 6 months (infants) to 75 and older. Nine studies had close to 100% female participants [24,25,38,42,43,46–48,52], and three studies had 100% male participants [23,29,53]. In six studies, gender distribution was either even or the difference between them was less than 10% [26,28,34,36,44,49]. In total, 30 studies aimed to reduce animal-derived foods (mainly red and processed meat intake) due to health concerns (cancer, overweight/obesity, high risk of developing Type 2 diabetes, ischemic heart disease), and only three studies considered both health and the environmental concerns in reducing animal-derived foods [23,26,27]. Half of the studies (n = 17) used behavioural change theories to guide their interventions. The Social Cognitive Theory (SCT) was the most frequently used theory [31,34,39,47,49,50,55], following by the Theory of Planned Behaviour (TPB) [26,27] and the Transtheoretical Model (TTM) [32,38].

Of the 51 included studies, 18 studies were categorised as micro-environmental/structural level studies which were conducted in high income countries. Micro-environmental/structural level studies refer to those studies which aimed to change immediate food environments in which people make food choices. They included nine RCTs [56–64], two CTs [65,66], three field experimental design studies [67,68,73], one quasi-experimental design study [69], and three pre-post design studies [70–72]. The number of participants ranged from 24 to 3,066 participants, and the participants' age ranged from 12 to 75 years and older. In eight studies, there were no significant differences in gender ratio of the sample (less than 10%) [56–58,62,63,66,69,70] and six studies did not provide the sex differences between the participants [59,60,65,67,71,72]. Nearly all studies took place in dining facilities, for example restaurants, cafés or worksite canteens; one study selected a farm and a small community as their participants [70]. In total, eight studies focused on health concerns only when designing the interventions to reduce unsustainable protein intake [60–62,64,66,68,71,72], two studies were developed to address the environmental considerations [65,73], and nine studies considered both health and environmental concerns for reducing unsustainable protein intake [56–59,63–65,69,70]. Twelve studies reported using one or more behavioural change theories in their intervention design [56–59,61,62,65,68–71,73]. Nudge theory either on its own or in combination with other theories (e.g. choice architecture, TPB) was used most commonly [56,58,59,65,68,69,73].

## Study quality

The overall methodological quality for all included studies was 'strong' for 12 studies, 'moderate' for 28 studies and 'weak' for 11 studies. The quality assessment for each individual study on each individual criterion is provided in Table 5.

## Individual level studies

**Educational interventions.** Twelve studies (RCT = 8, Pre/post = 4) used educational approach to reduce red/processed meat intake and purchase behaviour which included tailored education, educational classes, workshops and courses. Among RCT studies, six found positive impacts on the reduction in red meat intake in IG in comparison to CG

**Table 3. Characteristics of included individual level studies (n = 33).**

| Author(s) (year), country | Sample characteristics | Perspective | Intervention | Behaviour change theory | Outcomes | Results |
|---|---|---|---|---|---|---|
| **Randomised controlled Trials (RCT)** | | | | | | |
| Amiot et al., 2018 Canada | Adults Age(yrs): Mean = 23.5±3.1 Sample size: IG = 16 CG = 16 Male: 100% | Health and environment | The intervention comprised of 4-week multicomponent intervention that aimed to reduce meat intake including 5 components: a social norm; an informational/educational; an appeal to fear (animal farming/harm); a mind attribution induction; and a goal setting/self-monitoring. It also included 3 in-lab sessions for the intervention group and 2 for the control group. | Self-Determination Theory | Meat intake (red meat, white meat, fish and cold cuts; in grams) in total, during the week and weekend. Assessed using a dietary journal at baseline (T1), 2 weeks (T2) and 4 weeks (T3) post-intervention. | No significant changes between the groups were for total meat intake; total, week and weekend white meat intake; weekend red meat intake; weekend fish intake; and weekend cold cuts intake. The IG consumed significantly less red meat in total and at T3 compared to CG ($M_{diff}$ = 186.06, $p<0.025$); and during the week from T1 to T3 ($M_{diff}$ = 73.91, $p<0.025$). |
| Archarya et al. 2004 USA | Adults (Premenopausal women) Age(yrs): Mean(IG) = 43.2 ±2.7 Mean(CG) = 42.8±2.8 Sample size: IG = 100, CG = 106 Female: 100% | Health | The intervention aimed to increase soya intake among premenopausal women. The IG was given approx. 50mg of isoflavones per day for 2 years. All participants were counselled by a registered dietitian to learn how to incorporate soya products into their regular diet. | | Red/processed meat and soya intake (serves). Assessed using FFQ and 24-h recalls. | IG consumed significantly more soyabean products in comparison to CG from baseline to 1-year intervention (Mean±SD): IG 0.1±0.1 to 1.8 ±0.6; CG 0.1±0.2 to 0.2±0.3. Also, IG reduced intake of red meat but not significantly: IG 1.2±1.0 to 1.1± 0.9; CG 1.1 ±0.9 to 1.1±0.9. |
| Beresford et al., 2006 USA | Adults (Postmenopausal women) Age(yrs): 50 to 79 Sample size: IG = 19,541, CG = 29,294 Female: 100% | Health | The intervention was an intensive behavioural modification program that aimed to reduce dietary fat and meat intake. It involved 18 group sessions in the first year and quarterly sessions thereafter led by specially trained and certified nutritionists. Each participant was given their own dietary fat-gram goal according to the height. It emphasized self-monitoring techniques and introduced other tailored and targeted strategies. CG: received a copy of the US Department of Health and Human Services' Dietary Guidelines for Americans and other health related materials but were not asked to make dietary changes. | | Red meat intake (serves/d). Measured at baseline and at year 3 follow up using FFQ and 4-day food record. | The intervention was associated with statistically significant reduction in red meat intake (IG = Mean -9.7 ±128.4; CG = Mean 10.5±114.; $Mean_{Diff}$ = -20.2 (-25.5 to -14.8). |
| Carfora et al., 2017a Italy | Adults (Undergraduate students) Age(yrs): Mean = 19.37 ±1.55 Sample size: IG = 55, CG = 57 Male: 44% | Health and environment | The intervention aimed to reduce processed meat consumption (PMC) in young adults and consisted of a combination of encouragement of written self-monitoring of behaviour and anticipated regret as behaviour change techniques. IG received a daily SMS for 1 week on *Whats App*, which focused on anticipated regret and urged them to self-monitor PMC. CG: no intervention. | Theory of Planned Behaviour | PMC (portion/week). Measured with an online food diary. Questionnaire measured intensions, affective and instrumental attitudes and anticipated regret on PMC. | The IG had statistically significant changes in reduction of PMC at T2 ($F_{(1,112)}$ = 13.09; $p< 0.001$, np2 = 0.11). Also, intervention showed significant effects on instrumental attitude ($F_{(1,112)}$ = 8.81; $p<0.004$, $np^2$ = 0.09), anticipated regret ($F_{(1,112)}$ = 5.40; $p < 0.02$, $np^2$ = 0.06) and intentions ($F_{(1,112)}$ = 7.32; $p< 0.008$, $np^2$ = 0.06). |

*(Continued)*

**Table 3.** (Continued)

| Author(s) (year), country | Sample characteristics | Perspective | Intervention | Behaviour change theory | Outcomes | Results |
|---|---|---|---|---|---|---|
| **Carfora et al., 2017b Italy** | Adults (Undergraduate students) Age(yrs): Mean(IG) = 19.29±1.75 Mean(CG) = 19.29±1.04 Sample size: IG = 116, CG = 112 Male: IG = 28%, CG = 29% | Health and environment | The intervention aimed to reduce red meat consumption (RMC) in young adults. It used text messaging interventions to decrease RMC. IG received a daily SMS, which focused on anticipated regret and urged them to self-monitor RMC. CG: no intervention. | Theory of Planned Behaviour | RMC (portion/week). Measured with an online food diary. Questionnaire to measure intensions, affective and instrumental attitudes, subjective norms on RMC. | The intervention was effective in increasing intentions and reducing RMC. Results showed significant effects of condition for intention (F $(1,226) = 9.36$; $p < 0.01$, $n^2 = 0.04$), perceived behavioural control ($F(1,226) = 5.14$; $p < 0.05$, $n^2 = 0.02$), instrumental attitude (F $(1,226) = 23.84$; $p < 0.001$, $n^2 = 0.10$), healthy-eating identity ($F(1,226) = 11.08$; $p < 0.001$, $n^2 = 0.05$), and weekly RMC at T2 ($F(1,226) = 29.76$; $p < 0.001$, $n^2 = 0.12$). |
| **Celis-Morales et al., 2017 Ireland, The Netherland, Spain, Greece, UK, Poland and Germany** | Adults Age(yrs): Mean = 39.8 Sample size: IG1 = 312, IG2 = 324, IG3 = 321, CG = 312 Male: 41% | Health | The intervention aimed to evaluate the effect of different levels of personalised nutritional (PN) advice on intakes of major food groups including red meat. The Food4Me four-arm RCT conducted across 7 European countries. Participants were randomized to a CG (Level 0) or to one of three PN IG with increasingly more detailed personalized dietary advice (Levels 1–3) for a 6-month period. | | Red meat intake (g/d). Assessed at baseline and 6-month follow-up using online questionnaire. | Individuals receiving PN advice consumed less red meat (8.5%) at follow up. Red meat (g/day) for IG = 59.3 and CG = 64.7 (-5.48 (-10.8 to -0.09), $p = 0.046$. |
| **Carmody et al., 2008 USA** | Adults (Patients with prostate-specific antigen level and their partners) Age(yrs): Mean = 69.1±9.0 Sample size: IG = 17, CG = 19 Male: 100% | Health | The intervention was 11 weekly 2.5-hour classes that aimed to reduce meat intake. IG classes integrated didactic and experiential components on learning to shop for, and cook meals compliant with the study diet, and the use of mindfulness as a support in the dietary change. The participants received a study manual with background nutritional information and a cookbook of study-compliant meals and cooked and ate together a study-compliant meal at each class. CG: received the usual care, with the option of the intervention after their study participation. | | Red meat intake (g). Assessed at baseline, immediately after and 3 months after intervention using the 24-Hour Dietary Recall Nutrition Data System. | The IG showed significant reductions in: animal proteins (g)- baseline: 48±19, immediately after intervention: 28±17.1, 3 months after intervention: 28 ±16.3, $p = 0.03$; and vegetable protein (g): baseline: 27±8.4, immediately after intervention: 39±14.8, 3 months after intervention: 43 ±15.1, $p = 0.0002$. No significant changes in the CG. |
| **Dalgard et al., 2001 Denmark** | Adults (Patients with ischemic heart disease) Age(yrs): Mean(IG) = 55.5 ±11.3 Mean(CG) = 56.2 ±8.2 Sample size: IG = 17, CG = 19 Male: N(IG) = 15, N(CG) = 16 | Health | The intervention aimed to change dietary habits (including reducing meat intake) of patients 1 year after they received dietary advice. IG received either dietary advice on using the Plate Model and how to increase intakes of fruits and vegetables in a 10-minute session and CG received dietary advice primarily based on the National Cholesterol Education Program provided in 2 individually tailored 50-min session held 3 months apart. | | Meat intake (g/day). Assessed using food records measured at baseline, 12 weeks and 52 weeks after counselling. | The comprehensive counselling group significantly reduced meat intake: difference between groups from baseline to Week 52: 34(6;61), $p = 0.01$. |

*(Continued)*

**Table 3.** (Continued)

| Author(s) (year), country | Sample characteristics | Perspective | Intervention | Behaviour change theory | Outcomes | Results |
|---|---|---|---|---|---|---|
| **Delichatsios et al, 2001a USA** | Adults Age(yrs): Mean(IG) = 49.9 ±12.5 Mean(CG) = 56.8 ±12.9 Sample size: IG = 195, CG = 252 Male: 30% | Health | The intervention aimed to improve dietary habits among adult primary care patients. It comprised of mailed personalized dietary recommendations and educational booklets, verbal endorsement by the primary care provider; and 2 motivational counselling sessions with telephone counsellors. | Transtheoretical Model | Red/processed meat intake (serves/week). Assessed using FFQ at baseline and 3-month follow-up. | There was no intervention effect on red meat products in IG and CG. Red/processed meats: Adjusted difference (95% CI) = 0.0 (-0.3, 0.3). |
| **Delichatsios et al, 2001b USA** | Adults Age(yrs): Mean(IG) = 46.2 ±12.2 Mean(CG) = 45.7 ±12.5 Sample size: IG = 148, CG = 150 Male: 28% | Health | The intervention aimed to improve individuals' diet and was delivered via a totally automated, computer-based voice system. IG: the system monitored dietary habits and provided educational feedback, advice, and behavioural counselling. CG: received physical activity promotion counselling. | Social Cognitive Theory | Red/processed meat intake (serves/week). Assessed using FFQ from baseline, 3 and 6 months. | Changes in intakes of PMC and RMC showed trends in more healthful intake, although these trends were not statistically significant: Adjusted difference from baseline to 3 months (95% CI) = -0.1 (-0.3, 0.2) and from baseline to 6 months (95% CI) = -0.1(-0.3, 0.1). |
| **Emmons et al, 2005a USA** | Adults (Patients who resided in low-income, multiethnic neighbourhoods) Age(yrs): Mean(IG) = 50.8 Mean(CG) = 47.8 Sample size: IG = 1088, CG = 1131 Male: IG = 39.5%, CG = 29.1% | Health | The intervention aimed to change diet and comprised of: (1) study endorsement from the participant's clinician at a scheduled routine care visit, including provision of a tailored prescription for the recommended health behaviour changes; (2) an initial in-person counselling session with a health adviser; (3) 4 follow-up telephone counselling sessions with the health adviser; sets of tailored materials written for low-literacy audiences that targeted social contextual factors and links to relevant local activities. | | Red meat intake (serves/week). Assessed using FFQ measured at baseline and at 8 months. | Significantly greater change was found among participants in IG in red meat intake (≤3 servings per week) ($p <$ .001). 12% of the IG reduced red meat intake to ≤ 3 servings per week, compared with no change in the CG. |
| **Emmons et al, 2005b USA** | Adults (Patients who had undergone either sigmoidoscopy or colonoscopy) Age(yrs): 40–60+ Sample size: IG = 591, CG = 656 Male: 58% | Health | The project PREVENT aimed to change in the multiple risk factors that pose risk for colorectal cancer and other cancer development. IG received (a) a motivational and goal-setting telephone session; (b) 4 follow-up telephone counselling calls at monthly intervals. CG: received usual care. | Social Cognitive Theory | Red meat intake (serves/week). Assessed using FFQ. | IG experienced significantly greater improvement in reduction in weekly servings of red meat <3 servings per week than the CG ($p = 0.002$). |
| **Grimmett, et al, 2015 UK** | Adults (Colorectal cancer survivors) Age(yrs): Mean = 65 Sample size: 29 Male: 38% | Health | The intervention targeted physical activity, and intakes of fruit, vegetable, red and processed meat. It comprised 2-weekly telephone consultations with researcher for 12 weeks and supporting resources including meat-free menus. CG: Social support was encouraged. | | Red/processed meat (RPM) intake (g/week). Assessed using a modified version of the Health Education Authority FFQ. | RPM intake decreased from pre- to post-intervention (Mean reduction for red meat: 147.4, $p = 0.013$; mean reduction for processed meat: 0.83, $p = 0.002$). |

(*Continued*)

**Table 3.** (*Continued*)

| Author(s) (year), country | Sample characteristics | Perspective | Intervention | Behaviour change theory | Outcomes | Results |
|---|---|---|---|---|---|---|
| **Hatami et al, 2018 Iran** | Adults Age(yrs): +50 Sample size: IG = 48, CG = 50 Male: IG = 54%, CG = 50% | Health | The intervention targeted dietary changes using multimedia. The IG group received an audio-visual compact disc (CD) that contained information about nutritional behaviour of colorectal cancer prevention based on Health Belief Model (HBM) that lasted 45 min. | Health Belief Model | Red meat intake (serves). Measured using questionnaires and a 3-day dietary recall at the baseline, 1 week after, and 3 months after the intervention | There was a significant decrease in red meat servings ($p = 0.016$) in IG compared to the CG. In the CG, the results demonstrated a significant increase red meat intake ($p = 0.045$). |
| **Jaacks et al, 2014 USA** | Adults (At high risk of developing Type 2 diabetes) Age(yrs): 20% >65 Sample size: IG(lifestyle) = 1079, IG(metformin) = 1073, CG(placebo) = 1082. Female: ≥ 50% | Health | The intervention targeted dietary intake, with an emphasis on food groups. Participants were randomized to 3 groups (lifestyle intervention, metformin or placebo) for an average of 3 years. The lifestyle intervention involved a 16-session core curriculum over the first 24 weeks, followed by an individualized counselling curriculum (at least monthly contact). | | Red meat intake (serves). Assessed using a FFQ at baseline and at 1, 5, 6 and 9 years. | Participants in the lifestyle arm had significantly lower red meat intakes compared with the participants in the metformin and placebo arms. This change was statistically significant at 5 and 9 years ($p<0.05$). |
| **Johansen et al, 2009 Norway** | Adults (Women living in Norway and born in Pakistan or women born in Norway for 2 Pakistani parents) Age(yrs): Mean(IG) = 40.9 Mean(CG) = 41.5 Sample size: IG = 101; CG = 97 Female: 100% | Health | The intervention aimed to change dietary behaviour for Pakistani women living in Norway. IG received culturally adapted lifestyle education, including diet and physical activity. CG did not receive lifestyle advice except for the advice that they might have received by their GP or at the health-care centre. | Transtheoretical Model | Red meat intake (portions/week). Assessed using FFQ at baseline and after the 7-month intervention. | The daily intake of red meat was reduced in the IG ($P = 0.001$) but it was insignificant when comparing to CG ($p = 0.063$). |
| **James et al, 2015 Australia** | Adults (Cancer survivors and carers) Age(yrs): Mean(IG) = 56.2 ±12.6 Mean(CG) = 58.1±11.2 Sample size: IG = 75, CG = 58 Male: IG = 20%, CG = 26% | Health | The intervention targeted physical activity (PA) and diet. IG: face-to-face, group-based intervention (6 theory-based 2-hour sessions delivered over 8 weeks targeting healthy eating and PA). CG: Waitlist (after completion of 20-week data collection). | Social Cognitive Theory | Red/processed meat (RPM) intake (g/day). Assessed using FFQ at baseline, 8 and 20 weeks. | No significant changes in RPM intake between groups after 8 or 20 weeks. RMC: adjusted mean difference (95% CI) 8 weeks = -4.1(-28.3 to 20.1); 20 weeks = 6.8(-17.3 to 30.9), $p = 0.4208$. PMC: adjusted mean difference (95% CL) 8 weeks = 1.8(-3.2 to 6.7); 20 weeks = 3.1(-3.9 to 10.1), $p = 0.6659$. |
| **de Liz et al, 2018 Brazil** | Adults (Women undergoing breast cancer treatment) Age: NR Sample size: IG = 18, CG = 68 Female: 100% | Health | The intervention targeted diet including meat intake. IG participated in 12-month program by biweekly phone calls, personal meetings, and monthly handouts, while targeting the intake of at least 400 g/day of fruits and vegetables, and no more than 500 g/week of red or processed meats. | | Red/processed meat (RPM) intake (<500 g/week). Assessed using FFQ at baseline and after the intervention. | The IG improved their adherence and intake of RPM intake to the guidelines (<500g/week). Change in meat intake before and after the intervention IG = -219.6 (-778.1;-77.0). |
| **Lee et al, 2018 China** | Adults (Colorectal cancer survivors) Age(yrs): Mean = 65.2 Sample size: IG (Dietary and PA) = 55, IG(Dietary) = 56, IG(PA) = 56, CG = 56 Male: 63% | Health | The intervention targeted diet and PA. The interventions included individual face-to-face motivational interviews, fortnightly motivational phone calls, mailed monthly stage-of-change matched educational pamphlets, mailed quarterly newsletters, and quarterly group meetings. | | Red/processed meat (RPM) intake (serves/week). Assessed using FFQ measured at baseline and months 6, 12, 18, 24. | Dietary interventions significantly reduced RPM at all time-points (95% CI) = 0.88 (2.32 to 6.50), $p<0.001$. |

(*Continued*)

**Table 3.** (Continued)

| Author(s) (year), country | Sample characteristics | Perspective | Intervention | Behaviour change theory | Outcomes | Results |
|---|---|---|---|---|---|---|
| **Matthews et al, 2019 Finland** | Infants to 20-year-old adults Age: 13 months to 20 years Sample size: IG = 540, CG = 522 | Health | The intervention (STRIP) targeted dietary behaviour. IG received dietary counselling biannually from age 7 months to 20 years. CG did not receive any intervention. | | Red/processed meat (RPM) intake (g/day). Assessed using 4-day food records annually. | No difference in the intake of RPM (b: β = −1.19, 95% CI − 3.76–1.39, $p$ = 0.37). |
| **Merrill et al, 2009 USA** | Adults Age(yrs): Mean(IG) = 56.8 ±8.7 Mean(CG) = 58.0±9.0 Sample size: IG = 69; CG = 50 Female: 100% | Health | The plant-based dietary intervention targeted the intake of dairy products and meat. It included an intensive 40-hour educational course delivered over a 4-week period. Physical and dietary behaviours were promoted using health education and positive reinforcement. | Learning Theory (behaviourism) | Meat intake (serves/day). Assessed using the dietary questionnaire administered at baseline, 6 weeks, and 6 months. | After 6 months, the IG showed significant decreases in daily meat intake. Difference in means between baseline and 6 weeks in IG = -0.3 (95% CI = -0.5,-0.1). Difference in means between baseline and 6 months in IG = -0.5 (95% CI = -0.7,-0.3). |
| **Saffari et al, 2014 Iran** | Adults (Obese/overweight women) Age(yrs): Mean(IG) = 33.9 ±6.49 Mean(CG) = 34.62 ±5.63 Sample size: IG = 157, CG = 170 Female: 100% | Health | The intervention targeted dietary habits. IG sessions were carried out on a one-to-one basis with the implementation of 5 60-minute face-to-face sessions in the health centres. Motivational Interviewing (MI) techniques was used to encourage the participant's involvement, confirmation of the positive statements, using reflection to promote positive thinking. | | Meat intake (g/day). Assessed using FFQ measured at baseline and at months 3, 6, 9 and 12. | The intake of meat was significantly reduced in IG after intervention (P<0.05). IG: meat (g/d) change from baseline to one year after MI = -12.08 and CG: meat (g/d) change = -1.09. |
| **Sacerdote et al, 2005 Italy** | Adults Age(yrs): Mean(IG) = 44.7 ±12.6 Mean(CG) = 44.2 ±12.1 Sample size: IG = 1,592; CG = 1587 Male: IG = 50%; CG = 50% | Health | The intervention targeted dietary changes which included a non-structured 15-min educational intervention by general practitioners (GPs) on modifications of daily diet among healthy adults. IG: at the first visit the GP administered a 15-min personalized nutritional intervention. It focused on higher intake of fruits, vegetables, fish, and olive oil and lower intake of red meat, snacks, and sweets. CG received 'sham' intervention, which is a simpler and non-personalized conversation without the use of a brochure. | | Red meat intake (portions/ week). Assessed using FFQ measured at baseline and at 1 year of follow-up. | IG showed a slightly reduced net intake of meat. The net change of meat intake at 1 year in the IG was -0.22 (CI -0.11 to -0.69). |
| **Shai et al, 2012 Israel** | Adults (Health care providers (HCP) and patients) Age(yrs): Mean(HCP/IG) = 43.5±13.9 M(HCP/CG) = 48.6±11.3 Mean (patients/IG) = 34.5±9.1 Mean(patients/CG) = 35.8 ±9.1 Sample size: HCP(IG) = 55, HCP(CG) = 22, patients(IG) = 346, patients(CG) = 150 | Health | The Promoting Health by Self Experience (PHASE) intervention was multidisciplinary lifestyle intervention which comprised of 5 workshop days over 3 months in a small group sessions. | Theories of experimental learning and Bridges' model of change | Red meat intake. Assessed using questionnaire measured at baseline and after 3–4 months. | Among patients in IG, there was an overall improvement in dietary patterns, with decrease in red meat intake ($p$<0.05). |

*(Continued)*

**Table 3.** (Continued)

| Author(s) (year), country | Sample characteristics | Perspective | Intervention | Behaviour change theory | Outcomes | Results |
|---|---|---|---|---|---|---|
| **Zuniga et al, 2018 USA** | Adults (Overweight and obese, early-stage breast cancer survivors (BCS)) Age: M(IG) = 55.3 (SD 10.3), M(CG) = 58.4 (SD 8.2). Sample size: IG = 76, CG = 77. Female: 100% | Health | The intervention aimed to increase adherence to a Mediterranean style, anti-inflammatory dietary pattern in BCS. In the 6-month intervention, IG received monthly nutrition and cooking workshops, motivational interviewing telephone calls, and individualized newsletters. CG: received monthly informational brochures and no navigational services. | | Red meat intake (serves/day). Assessed using questionnaire measured at baseline and 6 months. | IG significantly reduced red meat intake to < 1 serving/day. At 6 months, 86.4% of the IG reported consuming less than one serving of red meat a day ($p = 0.002$). |
| **Non-randomised controlled Trials (CT)** | | | | | | |
| **Schiavon et al, 2014 Brazil** | Adults (Women with breast cancer) Age(yrs): Mean = 51 Sample size: IG = 18, CG = 75 Female: 100% | Health | The 12-month intervention targeted nutritional factors (red/processed meat and fruit and vegetable intake) and oxidative stress during treatment of breast cancer. It comprised of information bi-weekly phone calls, bi-monthly 24-hour dietary recalls followed by researchers' feedback and supporting materials. CG received basic healthy lifestyle guidelines at baseline and follow up. | Social Cognitive Theory | Red/processed (RPM) meat intake (g). Assessed using FFQ at the baseline and after 12 months. | A significant reduction in RPM consumption was observed between the groups in unadjusted analyses (B(exp) = 0.5, $p < 0.05$). |
| **Pre-post design** | | | | | | |
| **Hawkes et al- 2009 Australia** | Adults (Patients who had undergone surgery or chemotherapy) Age(yrs): Median = 66.0 Sample size: 20 Male: 50% | Health | The intervention (CanChange) was a 6-week telephone-delivered intervention by health coaches and supported by an interactive participant handbook. It targeted intake of red/processed meat, fruit and vegetable, alcohol, weight management, PA and smoking. It included lifestyle support, health risks information, behaviour change strategies, self-efficacy, and outcome expectations. | Social Cognitive Theory | Red/processed meat (RPM) intake (serves/week). Assessed using survey at baseline and post intervention (6 weeks follow up). | There was a significant decrease in RPM intake of from baseline (Median = 1) to post intervention (Median = 0, $p = 0.01$). No changes for red meat intake pre- and post- intervention. |
| **Hawkes et al, 2012 Australia** | Adults (First-degree relatives of colorectal cancer survivors) Age(yrs): Mean = 47.3 ±13.4 Sample size: 22 Male: 18% | Health | The intervention targeted behavioural risk factors for colorectal cancer including PA, diet (red/processed meat intake, fruit and vegetable intake), alcohol, weight management and smoking). The intervention included 6 x 1hour telephone health coaching sessions focus on motivation, expectations, values, mindfulness, expectations, action planning, goal-setting, self-monitoring, and a participant handbook and a pedometer. | Social Cognitive Theory | Red/processed meat intake (serves/week). Assessed using questionnaire at baseline and 6 weeks post intervention. | Processed meat intake decreased pre- to post-intervention (mean change, 95%CI = -1.2, -1.8 to -0.5, $p < 0.01$). No changes for red meat intake pre- and post-intervention. |

*(Continued)*

**Table 3.** (Continued)

| Author(s) (year), country | Sample characteristics | Perspective | Intervention | Behaviour change theory | Outcomes | Results |
|---|---|---|---|---|---|---|
| **Flynn et al, 2013** USA | Adults (Food Pantry clients) Age(yrs): Mean = 51.8 ±16.6 Sample size: 63 Male: 16% | Health | The intervention aimed to improve the food purchases of food pantry clients while decreasing food expenditures. It consisted of a 6-week cooking program which included plant-based recipes with a goal that participants would use the recipes for 3 meals per week. The cooking classes lasted about 30 min and involved a demonstration of one of the recipes. | | Meatless meals and meat purchasing behaviour. Assessed using a questionnaire at baseline, after 4 prior to the cooking program, 6 weeks of cooking, and after 6 months. | Grocery receipts showed a decrease in purchases of meat (p< 0.01). Average dollars/week spent on meat decreased significantly at baseline to follow-up (16.45±2.20 to 7.54 ±0.71; $P<0.001$). The number of meals per week that were plant-based recipes and did not contain meat/poultry/seafood increased significantly from baseline to follow-up (0.6±1.1 vs 2.8±1.3; $P< 0.01$). |
| **Lessem et al- 2019** USA | Adults (Nurse practitioners) Age(yrs): 25 to 65+ Sample size: 30 Female: N = 29 | Health | The intervention aimed to increase health care providers' knowledge and acceptance of whole-food plant-based (WFPB) diets and increase their likelihood of counselling patients on this dietary pattern. It was an online program comprising of a daily meal plan, shopping lists, education and motivational information. Participants received a weekly email with an educational voiceover PowerPoint presentation and WFPB information and resources. | Pender's health promotion model | Meat and legume intake (serves/week) Assessed using FFQ at baseline and post intervention. | Participants decreased intake of animal-derived foods. The largest change was in a 174% increase in legume serving (pre-4.43 to post- 12.13 and an 86% decline in meat servings per week (pre-intervention 8.57, post-intervention 1.2). |
| **Maryuyama et al, 2017 Japan** | Adults Age(yrs): 30–49 Sample size: 33 Male: 100% | Health | The intervention targeted metabolic risk factors for atherosclerotic cardiovascular disease in middle-aged men. Participants attended a one-hour nutrition education class to learn food items and recommended volumes comprising the Japan Diet, and were encouraged to consume the Japan Diet for 6 weeks. | | Meat and poultry intake (g). Assessed using a 3-day weighted dietary records at baseline and post intervention. | Intakes of meat and poultry decreased significantly post intervention (P = 0.011) from baseline (g) = 134.3 ±52.3 to 6 weeks = 95.4±73.0. |
| **Ring et al, 2019** USA | Adults Age(yrs): Mean(cohort 1) = 22.44 ±0.88 Mean(cohort 2) = 22.08±1.08 Sample size: cohort 1 = 9, cohort 2 = 12 Female: cohort 1 = 6, cohort 2 = 8 | Health | The intervention (*Cooking Up Health* (CUH) aimed to improve medical students' cooking and nutrition confidence, attitudes, and behaviours. CUH culinary elective module was developed to students and included: combination of lectures and readings, group meetings for interactive in counselling and motivational interviewing to promote healthy behaviours. Each class concluded with a hands-on chef-led culinary session on preparing plant-based meals along with a group dinner. | | Meat intake (servings). Assessed using PrimeScreen Dietary Screening Tool measured at baseline and post intervention. | Participation in CUH elective was associated with decreased meat consumption over time for students in cohort 1 ($p$ = 0.045, d = 1.49); and cohort 2 also showed decrease in red meat intake but effect was marginal in statistical significance (p = 0.08, d = 1.55). |

*(Continued)*

**Table 3.** (Continued)

| Author(s) (year), country | Sample characteristics | Perspective | Intervention | Behaviour change theory | Outcomes | Results |
|---|---|---|---|---|---|---|
| **Spees et al, 2016** USA | Adults (Cancer survivors) Age(yrs): Mean = 62 Sample size: 22 Male: 23% | Health | The intervention was a 4-month multifaceted intervention focusing on nutrition, PA and behavioural modifications delivered within a garden setting. It included harvesting produce >3 times a week, biweekly group education classes, access to remote motivational interviewing caching by a trained registered dietitian nutritionist, access to a secure online web portal for lifestyle behaviour recommendations and wellness tips, recipes and other resources. | Social Cognitive Theory | Red/processed meat intake (unit). Assessed using dietary screener questionnaire at baseline and immediately post-intervention. | The intervention resulted in significant decreased consumption of red and processed meats ($p = 0.030$). |

IG-intervention group, CG- control group; FFQ- Food Frequency Questionnaire; PA-physical activity; NR- not reported; PMC- process meat consumption RMC- red meat consumption.

[25,29,36,43,45,46], follow up varied between 3 months to 6 years. Two RCTs found that daily consumption of red meat was reduced in the IG, but it was not significant in comparison to CG [38,39]. Among pre-post studies: one study found a significant decrease in processed meat intake from baseline to 6 weeks [49]; one study found decrease in meat servings per week by 86% over 3 weeks period and significant increase in legume intake (pre-intervention 4.43 servings to 12.13 servings post-intervention) [52]; one study showed that intakes of red meat and poultry decreased significantly post intervention from baseline to 6 weeks [53]. Finally, one study found a decrease in purchases of meat (average dollars/week spent on meat) at baseline to 6 weeks [51].

**Counselling interventions.** Eleven studies (RCT = 9, CT = 1, Pre/post = 1) used counselling approach to reduce red/processed meat intake (n = 10) and increase in soya intake (n = 1). These interventions included telephone and in person counselling sessions providing dietary advice. Among RCT studies, nine studies found positive impact on reduction in red/processed meat intake in IG in comparison to CG [24,28,30,33–35,40,42,44] and increase intake in soyabean products [24], follow up period varied between 4 weeks to 24 months. One CT study showed that the IG improved their adherence of red/processed meat intake to the guidelines over 12-month program [48]. Pre-post study found that processed meat intake decreased pre- to post- intervention (6 weeks) but no changes for red meat intake were observed [50].

**Self-monitoring interventions.** Two studies used self-monitoring approach to reduce red meat intake. They were both RCTs and used daily text-messaging (SMS) approach to reduce red and processed meat intake. These studies urged participants to self-monitor meat intake and measured attitudes, intentions and anticipated regret [26,27]. It found positive impact on reduction in processed meat intake in IG in comparison to CG after one week [26], and red meat intake in IG in comparison to CG after two weeks [27].

**Multicomponent interventions.** Eight studies (RCT = 5, CT = 1, Pre/post = 2) used multicomponent approach to reduce red/processed meat intake such as education and self-monitoring [23], education and counselling approaches [31,32,37,41,47,54,55]. Among RCTs, three studies found no significant intervention effect on red meat/processed meat [31,32,41] and two studies showed significant reductions in red meat intake at 4 weeks [23] and over 9 years [37]. Also, a CT study showed significant reductions in red/processed meat intake among

**Table 4. Characteristics of included micro-environment studies (n = 18).**

| Author(s) (year), country | Sample characteristics & comparison | Perspective | Intervention | Behaviour change theory | Outcomes | Results |
|---|---|---|---|---|---|---|
| **Randomised Controlled Trials (RCT)** | | | | | | |
| Attwood et al. (2020), UK | Adults<br>**Study 1**<br>Age(yrs): 18 to 65+<br>Sample size: 147<br>Male: 39.5%<br>**Study 2**<br>Age(yrs): 18 to 65+<br>Sample size: 452<br>Male: 42% | Health and environment | This study investigated menu-based nudges on encouraging diners to move away from selecting meat-based meals and towards plant-based alternatives when choosing from food menus across different menu scenarios. Participants were randomly allocated to either a control (decoy absent) or intervention (decoy present) group. Participants were asked to choose between three dishes– a 'competitor' meat option, a 'target' vegetarian option and a 'decoy" vegetarian option. | Nudge theory | Choice of vegetarian option across different menu scenarios using online survey. | There was no significant effect of the intervention in study 1: decoy absent-(CG) vs. decoy present (IG): OR 0.50, 95% CI 0.22 to 1.15; p = 0.1.<br>In study 2 across 7 menu conditions and testing a more expensive decoy also showed no effect of the intervention decoy absent vs. decoy present: OR 0.68 (95% CI 0.41 to 1.12). |
| Bacon et al. (2018a), UK | Adults<br>Setting: Restaurant<br>Age(yrs): Median = 34<br>Sample size: 750; CG = 194<br>IG(recommendation menu) = 185<br>IG(descriptive menu) = 185<br>IG(vegetarian menu) = 186<br>Male: 47% | Health and environment | In an online scenario, participants were randomly assigned to 4 different restaurant menu conditions: control (all dishes presented in the same manner), recommendation (vegetarian dish presented as chef's recommendation), descriptive (more appealing description of vegetarian dish), and vegetarian (vegetarian dishes placed in a separate section). Participants were asked to imagine a scenario in which they were catching up with a friend for dinner and asked to choose a meal from allocated menu. Vegetarians and vegans were excluded. | Theory of Planned Behaviour | Frequency of choosing vegetarian dishes. | The recommendation menu (OR 1.1, 95% CI [0.618, 1.973] and the descriptive menu (OR 0.917, 95% CI [0.503, 1.673] did not influence vegetarian food choice in comparison to control menu. Vegetarian menu decreased the odds of selecting a vegetarian dish (OR 0.406, 95% CI [0.195, 0.848], p = 0.016. |
| Campbell-Arvai et al. (2014), USA | Adults<br>Setting: Dining facilities on university campus<br>Sample size: 319<br>Male: 47% | Health and environment | The intervention was a menu choice experiment with undergraduate students over 2 weeks. Participants were given approx. 5 min to consult 1 of 8 randomly assigned menus and make their selection. A 2x2x2 factorial design was used in menu choice experiment varying: 1) IG menus included 5 appealing meat-free options and a range of non-vegetarian dishes and CG menus included 5 less appealing meat-free options and a range of non-vegetarian dishes; 2) IG menus– meat options were removed and repositioned on a board 3.5m away and CG menus contained a range of meat-free and meat-based options; 3) IG menus contained a range of meat-based options and meat-free options with a leaf symbol indicating that consuming less meat can help to reduce environmental impact and CG menus contained a range of meat-free and meat-based options. | Nudge theory | Selection of meat-free food options on menus measured using a survey. | Participants viewing IG menus with 5 appealing meat-free options had lower odds of selecting meat options than did those viewing CG menus (OR 0.49, 95% CI 0.36–0.66). Also, participants viewing IG menus with removed meat options and repositioned on a board had lower odds of selecting meat options than did those viewing CG menus (OR 0.24, 95% CI 0.18–0.36). However, the odds of selecting a meat-based dish did not differ between participants viewing the IG meat-free menus with a leaf and the CG menus (OR 0.92, 95% CI 0.69–1.2). |

*(Continued)*

**Table 4.** (Continued)

| Author(s) (year), country | Sample characteristics & comparison | Perspective | Intervention | Behaviour change theory | Outcomes | Results |
|---|---|---|---|---|---|---|
| **Gravert & Kurz (2019), Sweden** | Adults<br>Setting: Restaurant<br>Sample size: 1 restaurant | Health and environment | The intervention consisted of 2 weeks intervention and 1-week post-intervention. The back of the restaurant served as CG and received a menu listing 1 meat and 1 fish option and the sentence "Vegetarian meal available on request". The front area served as IG and customers received a menu listing 1 vegetarian and 1 fish option (no meat) but with the sentence "option of meat available". | Nudge theory | Sales of vegetarian (meat-free) dishes. | Only 2.5% of all dishes sold were vegetarian without the vegetarian option on the menu. The share of meat dishes during the intervention dropped from 47% to 34%, a reduction of 38% ($p < 0.01$). The vegetarian dishes increased from 3% to 9% on average, a 200% increase ($p < 0.01$). Of all dishes sold, 15% were vegetarian in the vegetarian area but only 3.5% were vegetarian in the meat area ($p < 0.01$). |
| **Herbert et al. (1993), USA** | Adults<br>Setting: Workplaces<br>Sample size: 16 worksites assigned to either IG or CG, questionnaire responses N = 2365 (baseline). | Health | The intervention focused on 8 discrete food-based eating pattern messages aimed to increase fruit, vegetable, high fibre cereals, whole grain breads and rice or pasta, potatoes and fried beans, peas or lentils, substitution of low-fat dairy products and removal of skin from chicken and trimming visible fat from meat or substitution of fish and poultry for other meat. It consisted of classes, taste tests, food demonstrations, labelling of recommended foods in cafeteria and bulletin board displays. Intervention was tailored for each worksite. | | Ground and processed meat intake (serves) assessed using self-administered health habits questionnaire (HHR) including FFQ measured at baseline and after intervention. | A significant decrease in ground and processed meats were observed among intervention companies ($p = 0.05$). |
| **Kongsbak et al. (2016), Denmark** | Adults<br>Setting: Ad libitum buffet<br>Age(yrs): Mean(IG) = 23.8 ±0.4 Mean(CG) = 24.4±0.4<br>Sample size: N(IG) = 33, N (CG) = 32<br>Male: 100% | Health | A single one-day lunch meal study was conducted in a FoodScape Laboratory where an Intelligent Buffet was used to register the exact weight of each meal component self-served by each participant. The choice architecture in the IG consisted of altering the serving sequence and serving fruit and vegetable components in 8 separate bowls. In CG, all salad components were served together. | Choice architecture | Selection of self-served meatballs in grams using radio frequency identification technology. | There was no significant difference in selection of meatballs between IG (mean 156.2, SD 71.1; $p = 0.078$ and CG (mean 194.6, SD 78.6) after adjusting for BMI, age, and selection of salad, pasta and bread. |
| **McClain et al. (2013), USA** | Adults<br>Setting: University dining halls<br>Age(yrs): 20<br>Sample size: 4 university dining halls; 92 IG and 2 CG; questionnaire responses N(IG) = 247, N(CG) = 278<br>Male: 47% | Health | The intervention was a 4-week multicomponent, point-of-selection marketing intervention. For the intervention, 4 prototypes were developed: (1) students were given menus upon entry into the dining hall to help them decide their meal; (2) a "dimsum" style vegetable cart was pushed throughout the dining hall; (3) prepared balanced meals were placed on display at the dining hall's entrance; (4) a Chef's Pick of the Day that included a portrait of the chef and a plated vegetarian meal were placed on display at the front entrance of the dining hall. CG: 4 weeks of business as usual. | Social Cognitive Theory Transtheoretical Model | Intake of fruits, vegetables and high-fat meat measured as servings per week assessed at baseline and after intervention using FFQ. | Students in the intervention dining halls consumed significantly less junk food and high-fat meat and increased their perceived importance of eating a healthful diet (more fruits and vegetable servings per week) relative to the CG. In the IG, high-fat meat intake reduced by 0.9 servings per week and in CG increased by 0.9 servings per week; $p = 0.04$. |

*(Continued)*

**Table 4.** (Continued)

| Author(s) (year), country | Sample characteristics & comparison | Perspective | Intervention | Behaviour change theory | Outcomes | Results |
|---|---|---|---|---|---|---|
| Reinders et al. (2017), Netherlands | Adults Setting: Restaurant Age(yrs): Mean = 48.6±17.5 Sample size: 1006; IG = 470, CG = 536 Male: 46% | Health and environment | Three restaurants were randomly assigned to a sequence of an intervention and control condition. In the intervention period, the vegetable portion sizes on the plates of main dishes were doubled (150g of vegetables instead of 75g) and the portion sizes of meat on the plates were reduced by an average of 12.5% for 6 weeks. In the control period, the portion sizes of the main dishes were maintained as usual for 6 weeks. | | Vegetable and meat intake measured by subtracting the grams of meat returned to the kitchen from the average grams of meat in each of the targeted dishes. | Vegetable consumption from plates was significantly higher during the intervention period (Mean = 115.5g) than during the control period (Mean = 61.7g). Total vegetable intake (including side dishes) was significantly higher during the intervention period (Mean = 178.0 g) than during the control period (Mean = 137.0g). Meat intake was significantly lower during the intervention period (Mean = 183.1g) than during the control period (Mean = 211.1g) $p<0.001$. |
| Sorensen et al. (2005), USA | Adults Setting: Worksites and small businesses Age(yrs): Mean(IG) = 44 Mean(CG) = 43 Sample size: N(IG) = 13 N (CG) = 13; questionnaires N (IG) = 807; N(CG) = 933 Male: 67% | Health | The worksite intervention was 18-month multicomponent intervention aiming to increase fruit and vegetable intake, physical activity and reduce red meat intake and smoking. It included policies aimed at offering healthful food options at company meetings, interactive activities, and education. | | Fruit, vegetable, and red meat intake measured in servings per week using FFQ at baseline and directly after intervention. | At follow-up, 22% of workers at intervention worksites were eating 5 servings of fruits and vegetables per day, compared with between 12% and 15% of workers and managers at control worksites. The change in percentage of participants eating ≤3 servings per week of red meat did not differ between intervention (+4.1%) and control group (+3%), $p = 0.72$. |
| **Non-randomised controlled Trials (CT)** | | | | | | |
| Kurz (2018), Sweden | Adults Setting: University restaurant Sample size: 2 university restaurants (IG = 1 and CG = 1) | Health and environment | The intervention tested if nudging can increase the consumption of vegetarian food. At the treated restaurant, the salience of the vegetarian option was increased by changing the menu order and enhancing the visibility of the vegetarian dish. The other restaurant served as a control. | Nudge theory | Daily sales on three main dishes (one of which vegetarian) at the baseline (first 9 weeks) and intervention period of 17 weeks. | The nudge increased the share of vegetarian lunches sold by on average 6% point, and that the treatment effect increased over time. The change in behaviour was partly persistent, as the share of vegetarian lunches sold remained 4% point higher after the intervention ended than before the experiment. |
| Polak et al. (2019), Israel | Adults and children Setting: Rural Kibbutz cafeteria Age(yrs): Mean(IG) = 55.3 Mean(CG) = 52.7 Sample size: N(IG) = 493 adults and 214 children; N (CG) = 487 adults and 206 children. Male: IG = 43%, CG = 47% | Health | It was a community-based culinary coaching programme (CCCP). It included 8x90 min coaching sessions with a community steering committee, 22h of kitchen staff training, 12h of pre-school staff training and 30h of education for diners. Control community received no intervention. | | Food purchases of legumes, wholegrain products, fruits, nuts, vegetables, MUFA/SFA, processed meats and fish measured before and 12 months after programme initiation. | Intervention cafeteria food improved significantly in all Mediterranean index categories except nuts (legumes, wholegrain products, fish, MUFA/SFA $p<0.0001$; fruits, vegetables $p<0.001$; processed meats $p = 0.004$), and in the proportion of ultra-processed and unprocessed or minimally processed foods categories of the NOVA classification (−22%, $p<0.001$ and +7%, $p< 0.001$, respectively), compared with the control community. |
| **Field experimental design** | | | | | | |

*(Continued)*

**Table 4.** (Continued)

| Author(s) (year), country | Sample characteristics & comparison | Perspective | Intervention | Behaviour change theory | Outcomes | Results |
|---|---|---|---|---|---|---|
| **Bacon et al. (2018b), UK** | Café customers<br>Setting: Cafés<br>Sample size: IG = 10 cafés, CG = 18 cafés | Environment | The intervention was an 8-week intervention aimed to increase plant-based dish sales by changing the language used to describe plant-based options on restaurant menus. The experiment was conducted in a chain of cafés within Sainsbury's grocery stores in the UK. Three vegetarian versions of meat-based dishes were selected for this experiment. | | Sales of plant-based breakfast and lunch dishes assessed at baseline and 8 weeks after dish names were changed. | Changing the breakfast dish "Meat-free Breakfast" name to the alternatives of "Garden Breakfast" (OR = 1.13, 95%CI 1.00 to 1.26, p = 0.04) and "Field-grown Breakfast" (OR = 1.19, 95%CI 1.05 to 1.35, p = 0.008) led to significant increases in target vegetarian dish sales compared to CG. Alternative names: "Feel Good Fry Up" (OR = 1.08, 95% CI 0.97 to 1.20, p = 0.149), "Triple Cheese and Slow Roasted Vegetable Lasagne" (OR = 0.99, 95%CI 0.87 to 1.13, p = 0.896), "Florentine Lasagne" (OR = 0.95, 95%CI 0.82 to 1.10, p = 0.504), "Better Sausages and Mash" (OR = 1.07, 95%CI 0.71 to 1.59, p = 0.757) did not lead to a significant difference in dish sales compared to CG. Two of the three alternative names: "Field-grown Sausages and Mash" (OR = 1.52, 95%CI 1.10 to 2.10, p = 0.012) and "Cumberland Spiced Veggie Sausages and Mash" (OR = 1.77, 95%CI 1.18 to 2.64, p = 0.005), were associated with significant increases in dish sales compared to CG. |
| **Friis et al. (2017), Denmark** | Adults<br>Setting: FoodScape Lab, University<br>Age(yrs): Mean(Priming) = 27.3±6.6 Mean(Default) = 25.9±7.1 Mean(Variety) = 26.3±6.5<br>Sample size: N(Priming) = 24 N(Default) = 33 N (Variety) = 31<br>Male: 30–42% | Health | The intervention included three experiments: priming, default and perceived variety. In the default arm, the salad was pre-portioned into a bowl containing 200g of vegetables. Priming arm tailored the environment to accommodate a green ambience of plants, green servings bowls and herbs in the dining area. In the perceived variety arm the pre-mixed salad were divided into each of its components, to increase the visual variety of vegetables, yet not providing an actual increase in items. Control arm: the food environment was not manipulated. | Choice architecture<br>Nudge theory | Vegetable intake (the difference in weighed intake (consumption minus wastage) measured in grams using web-based questionnaire and FFQ. | Both the priming condition and perceived variety decreased the meat-based meal component. For the vegetable intake priming had an age-related effect with an 8g/year increase for the group (p = 0.01). Comparing the effect size of the three nudges (presented as the difference between intervention and control in g) a significant difference was seen between priming and default, with a mean difference of 201g in total intake and 81g for total vegetable intake. |
| **Prusaczyk et al. (2021), USA** | Adults<br>Setting: Online<br>Age(yrs): Mean = 37.71 ±11.54<br>N = 562<br>Male: 53% | Health and environment | The intervention aimed to reduce the willingness to order a all-beef burger in favor of a beef-mushroom burger. Participants were assigned to either nudge (a beef-mushroom burger as a default option), education (information was provided for participants about animal-derived food impact on environment) or control conditions. | Nudge theory | Willingness to order beef burger measured using questionnaire. | Both the nudge and education conditions showed positive results in reducing the willingness to order the all-beef burger vs beef-mushroom burger in comparison to control condition. |
| **Quasi-experimental design** | | | | | | |
| **Dos Santos et al (2020), Denmark, France, Italy and United Kingdom** | Adolescents<br>Setting: Foodservice/ restaurant<br>Age(yrs): 12–19<br>Sample size: 360<br>Male: In Denmark, France and UK- 50%; Italy- 60% | Health and environment | The intervention aimed to influence adolescents to select a vegetable-based dish when this dish was described as "dish of the day" (IG) compared to CG when this strategy was not used. This experiment was implemented in restaurants in 4 European countries: Denmark, France, Italy and United Kingdom. | Theory of Planned Behaviour<br>Nudge theory | Selection of vegetable-based dish ("dish of the day") measured using questionnaire. | The nudging strategy (dish of the day) did not show a difference on the choice of the vegetable-based option (p = 0.80 for Denmark and France and p = 0.69 and p = 0.53 for Italy and UK, respectively). However, natural dimension of food choice questionnaire, social norms and attitudes towards vegetable nudging were all positively associated with the choice of the vegetable-based dish. Being male was negatively associated with choosing the vegetable- based dish. |

*(Continued)*

**Table 4.** (Continued)

| Author(s) (year), country | Sample characteristics & comparison | Perspective | Intervention | Behaviour change theory | Outcomes | Results |
|---|---|---|---|---|---|---|
| **Pre-post design** | | | | | | |
| **Craveiro et al (2019), Portugal** | Adults<br>Setting: Small-scale farms<br>Age(yrs): Mean(Farmers) = 44.5±10.7 Mean (Consumers) = 44.5±10.7<br>Sample size: 36 farmers and 294 consumers.<br>Male: Farmers = 42%, Consumers = 20% | Health and environment | PROVE was a Portuguese program that empowers small-scale farmers organised into local networks to directly commercialize baskets of locally produced fruits and vegetables to consumers. Farmers received training, handbook, an access to online platform to access consumers. PROVE consumers subscribe to receiving baskets of an agreed range of fruits and vegetables with average weight of 7 kg. | INter-sectoral Health and Environment Research for InnovaTion model | Fruits, vegetables and red meat intake measured using FFQ. | PROVE consumers were more likely to eat ≥5 portions of fruits and vegetables a day in comparison to the matched sample of Portuguese citizens (average odds ratio: 3.05, $p < 0.05$). Also generated an impact on the likelihood of consuming ≤2 portions of red meat a week (average odds ratio: 1.56, $p < 0.05$). |
| **Resnicow et al. (1992), USA** | Children<br>Setting: Elementary school<br>Sample size: 5 elementary schools<br>Longitudinal cohort (n =1,209/ Post-test only cohort (n = 3,066). | Health | The Know Your Body (KYB) program is a comprehensive skill-based school health education program. It included classroom curriculum, school-wide activities (peer leader training, student health committees, food tasting and health lectures), and environmental modifications (school cafeteria). | Social Learning Theory | Meat intake assessed using questionnaire at baseline and 3 years post-test. | Students in the post-test only cohort who had high implementation teachers showed significantly ($p < 0.05$) lower self-reported intake of meat and desserts, as well as higher health knowledge and self-reported intake of "heart healthy" foods and vegetables than comparison students. |
| **Sperber et al. (1996), Israel** | Adults and children<br>Setting: Kibbutz community<br>Sample size: 208 adults and 123 children. | Health | The intervention included food policy changes in the central Kibbutz kitchen, health education programs targeting all aged groups and health counselling for individual at risk of coronary artery disease. Meals were usually consumed in communal dining rooms and prepared by kitchen staff. A registered dietitian was hired for 2 days to plan menus and advise where to buy produce with the cooks. Residents also received health education via newsletter, personal letters and mass media messages. | | Food purchases measured by a questionnaire at baseline and after 2 years. | Meat intake rich in saturated fat and cholesterol dropped by 80%; red meat intake decreased by approximately 19%. The intake increased for fish (+19.2%), chicken meat (+11.4%) and vegetarian patties (+80%) increased. |

IG-intervention group, CG- control group, FFQ- Food Frequency Questionnaire, BMI- Body Mass Index, OR- odds ratio.

**Table 5. Quality assessment of included studies.**

| Author(s) (year), country | Selection bias | Design | Confounders | Blinding | Data collection methods | Withdrawals and drop-outs | Overall |
|---|---|---|---|---|---|---|---|
| Amiot et al, 2018, Canada | 2 | 1 | 2 | 2 | 3 | 1 | 2 |
| Archarya et al. 2004, USA | 2 | 1 | 1 | 2 | 2 | 3 | 2 |
| Attwood et al. (2020), UK | 2 | 1 | 1 | 2 | 3 | 2 | 2 |
| Bacon et al. (2018a), UK | 2 | 1 | 3 | 2 | 2 | 1 | 2 |
| Bacon et al. (2018b), UK | 2 | 1 | 2 | 2 | 2 | 3 | 2 |
| Beresford et al., 2006, USA | 2 | 1 | 3 | 2 | 3 | 1 | 3 |
| Campbell-Arvai et al. (2014), USA | 1 | 1 | 3 | 2 | 2 | 1 | 2 |
| Carfora et al., 2017a, Italy | 2 | 1 | 3 | 2 | 2 | 2 | 2 |
| Carfora et al., 2017b, Italy | 2 | 1 | 3 | 2 | 2 | 1 | 2 |
| Carmody et al., 2008, USA | 2 | 1 | 3 | 2 | 3 | 3 | 3 |
| Celis-Morales et al., 2017, Ireland, The Netherland, Spain, Greece, UK, Poland and Germany | 1 | 1 | 3 | 2 | 2 | 2 | 2 |
| Craveiro et al (2019), Portugal | 2 | 2 | NA | 2 | 2 | 3 | 2 |
| Dalgard et al., 2001, Denmark | 3 | 1 | 3 | 2 | 3 | 1 | 3 |
| Delichatsios et al, 2001a, USA | 1 | 1 | 1 | 2 | 2 | 3 | 2 |
| Delichatsios et al, 2001b, USA | 3 | 1 | 2 | 2 | 2 | 1 | 2 |
| Dos Santos et al (2020), Denmark, France, Italy and UK | 2 | 1 | 2 | 2 | 3 | 3 | 3 |
| Emmons et al, 2005a, USA | 2 | 1 | 1 | 2 | 2 | 1 | 1 |
| Emmons et al- 2005b, USA | 2 | 1 | 1 | 2 | 2 | 1 | 1 |
| Flynn et al, 2013, USA | 2 | 2 | NA | 2 | 3 | 2 | 2 |
| Friis et al. (2017), Denmark | 2 | 1 | 2 | 2 | 2 | 2 | 1 |
| Gravert & Kurz (2019), Sweden | 2 | 1 | 3 | 2 | 3 | 3 | 3 |
| Grimmett, et al, 2015, UK | 2 | 1 | 3 | 2 | 2 | 2 | 2 |
| Hatami et al, 2018, Iran | 2 | 1 | 3 | 2 | 1 | 1 | 2 |
| Hawkes et al- 2012, Australia | 2 | 2 | NA | 2 | 1 | 3 | 2 |
| Hawkes et al, 2009, Australia | 2 | 2 | NA | 2 | 1 | 1 | 1 |
| Herbert et al. (1993), USA | 2 | 1 | 1 | 2 | 2 | 2 | 1 |
| Jaacks et al, 2014, USA | 2 | 1 | 1 | 2 | 2 | 3 | 2 |
| Johansen et al, 2009, Norway | 2 | 1 | 2 | 2 | 2 | 3 | 2 |
| James et al, 2015, Australia | 3 | 1 | 1 | 2 | 1 | 2 | 2 |
| Kongsbak et al. (2016), Denmark | 2 | 1 | 1 | 2 | 2 | 3 | 2 |
| Kurz (2018), Sweden | 2 | 1 | 2 | 2 | 2 | NA | 1 |
| Lee et al, 2018, China | 2 | 1 | 1 | 2 | 2 | 1 | 1 |
| Lessem et al- 2019, USA | 3 | 2 | NA | 2 | 1 | 1 | 2 |
| de Liz et al, 2018, Brazil | 2 | 1 | 3 | 2 | 2 | 2 | 2 |
| Maryuyama et al, 2017, Japan | 2 | 2 | NA | 2 | 2 | 1 | 1 |
| Matthews et al, 2019, Finland | 2 | 1 | 3 | 2 | 2 | 3 | 3 |
| McClain et al. (2013), USA | 2 | 1 | 1 | 2 | 1 | 3 | 2 |
| Merrill et al, 2009, USA | 2 | 1 | 1 | 2 | 1 | 1 | 1 |
| Polak et al. (2019), Israel | 2 | 1 | 1 | 3 | 2 | 3 | 3 |
| Prusaczyk et al. (2021), USA | 1 | 1 | 3 | 3 | 2 | 3 | 3 |
| Reinders et al. (2017), Netherlands | 2 | 1 | 1 | 2 | 3 | NA | 2 |
| Resnicow et al, 1992, USA | 1 | 2 | NA | 2 | 2 | 3 | 2 |
| Ring et al, 2019, USA | 2 | 2 | NA | 2 | 1 | 1 | 1 |
| Saffari et al, 2014, Iran | 2 | 1 | 1 | 2 | 3 | 1 | 2 |
| Sacerdote et al, 2005, Italy | 1 | 1 | 3 | 1 | 2 | 1 | 2 |
| Schiavon et al, 2014, Brazil | 2 | 1 | 2 | 2 | 2 | 2 | 1 |

*(Continued)*

**Table 5.** (Continued)

| Author(s) (year), country | Selection bias | Design | Confounders | Blinding | Data collection methods | Withdrawals and drop-outs | Overall |
|---|---|---|---|---|---|---|---|
| Shai et al, 2012, Israel | 2 | 1 | 3 | 2 | 3 | 1 | 3 |
| Sorensen et al. (2005), USA | 3 | 1 | 1 | 2 | 2 | 1 | 2 |
| Spees et al, 2016, USA | 2 | 2 | NA | 2 | 2 | 1 | 1 |
| Sperber et al. (1996), Israel | 2 | 2 | NA | 2 | 3 | 3 | 3 |
| Zuniga et al, 2018, USA | 1 | 1 | 1 | 2 | 3 | 1 | 2 |

*Strong-1; Moderate-2; Weak-3.

**Overall: Strong-1 (no WEAK rating), Moderate- 2 (one WEAK rating), Weak- 3 (two or more WEAK ratings).

women with breast cancer after 12 months [47]. Two pre-post studies showed significant changes in decreasing red/processed meat intake among students and cancer survivors [54,55].

## Micro-environment level studies

**Menu manipulation interventions.** Of 18 micro-environmental level studies, seven studies (RCT = 4, CT = 1, Exp = 3) used menu manipulation approach in order to reduce meat options or increase choice/sale of plant-based meal options [56–59,65,67,69,73]. Menu manipulation included different approaches such as adding attractive meat free choices on the menu, adding specific symbols specifying that less meat intake can save the environment, increasing the visibility of plant-based options, and describing the plant-based option as a 'Dish of the day'. Of four RCTs, two did not show a difference on the choice of plant-based options in IG compared to CG [56,57], and two showed a positive impact on meat reduction behaviour by choosing more plant-based options in restaurants in IG compared to CG [58,59]. Also, a CT study showed increase in sales of plant-based lunches [65]. One experimental study revealed a significant changes on plant-based dish sales by changing the language to explain plant-based options on café/restaurant menus (e.g. replacing Meat-Free Breakfast with Garden Breakfast) [67]. However, the other experimental study found that the nudging strategy a 'Dish of the day' did not show a difference on the choice of the plant-based option among adolescents [69].

**Choice architecture interventions.** Three studies used choice architecture approach (RCT = 2, Exp = 1) which included dining environment manipulation such as altering the serving sequence of plant-based and meat-based dishes [61], altering portion sizes of plant and meat-based foods [63] and using priming (environmental changes- adding green plants, herbs and green colour bowls), default (pre-portioned salad bowls) and perceived variety options (pre-mixed salad to increase the visual variety of vegetables) [68]. The manipulation of altering the serving sequence and default approach found no significant difference in selection of meat dishes between IG and CG [61,68]. However, the manipulation of altering portion sizes resulted in significant higher vegetable intake and lower meat intake in IG than CG [63] as well as using the priming and perceived variety conditions showed decrease in choosing meat-based options [68].

**Multicomponent interventions.** Seven studies (RCT = 3, CT = 1, Pre/post = 3) used multicomponent approach to reduce red meat intake which included combination of education, labelling, policy, counselling and choice architecture. Of three RCTs, two studies found significant decrease in ground and processed meat intake [60] and high-fat meat intake [62]. One RCT showed that percentage of participants eating ≤3 servings per week of red meat did not differ between IG and CG over 18-months [64]. One CT study and

three pre-/post- studies showed significant reduction in processed meat intake [66] and red meat intake [70–72].

## Discussion

This systematic literature review explored the effectiveness of interventions which aimed to promote protein consumption from low ecological footprint sources (reduction in animal-derived proteins and increase in plant-based proteins). Most of individual level studies demonstrated reduction of animal-derived protein intake, mainly measured by reduction in red and/or processed meat intake with only a few studies measured increase intake in plant-based proteins such as legumes and soyabeans. Furthermore, 52% of these studies (n = 17) targeted people with current or past chronic conditions such as cancer, diabetes and cardiovascular diseases. The majority of micro-environmental/structural level studies found positive dietary changes in reducing animal-derived protein intake mainly red meat and increase in plant-based protein (e.g. plant-based vegetarian dishes). The majority of these studies were conducted in dining facilities such as cafes, restaurants and canteens. Similar findings have been observed in a several systematic literature reviews which evaluated the effectiveness of interventions targeting to reduce demand for meat [74,75].

Educational, counselling and self-monitoring interventions are promising approaches to dietary behaviour change such as increase in environmentally sustainable protein intake which have been observed in other systematic literature reviews [76,77]. However, these approaches have been often used among at risk or highly motivated populations such as people with obesity or other chronic condition, cancer survivors, and may have limited success in changing behaviour among general population. In addition, research indicates that educational interventions on its own may not be sufficient to behaviour change in long term [77,78]. There is a need for further longitudinal studies to confirm that the reduction in animal-derived protein intake from high ecological footprint sources among healthy general population sustain over prolonged period of time.

The majority of our included interventions focused on reduction of red/processed meat intake mainly from a health perspective with only three studies emphasising the reduction of animal-derived foods due to negative impact on the environment. This is not a surprising finding as recent studies reported general population having low food literacy and limited understanding of food impact on the environment and often focus on changing their dietary behaviours such as reducing red/processed meat intake due to health reasons [79–82]. Therefore, there is a need to develop interventions aiming to educate general public on sustainable and healthy diets and evaluate how feasible these interventions are in reducing overconsumption of protein from high ecological footprint sources in order to reduce the negative impact it has on health and environment.

The findings from included micro-environmental behaviour change interventions showed that altering food environments using nudges or choice architecture can lead to positive dietary behaviour changes such as reduction in unsustainable protein intake/purchase and increase in plant-based meals which aligns with findings from previous studies [75,83]. Most promising approaches included altering portion sizes, menu manipulation by adding plant-based meal options and policy implementation; similar environmental approaches have been identified as successful in changing dietary behaviours among young adults [84]. In order to reduce the human behaviour towards more environmentally sustainable protein intake, it is important not only to change the supply but also the demand of unsustainable foods [78]. Interestingly, the most recent qualitative study indicated that young Australians were open to or interested in affordable meat alternatives such as plant-based meals and reported that often

these options were not available or very limited when dining out (unpublished work). This indicates that people may be interested in changing their dietary behaviours to more sustainable and healthy but food environments need to be supportive in helping them to make informed food choices. Choice architecture, nudging strategies and policy implementation can be promising approaches to create enabling food environments and for changing dietary behaviour towards more sustainable diets. However, there is still a need to develop and test different strategies among more general population and settings to determine what motivates them in choosing more environmentally sustainable food options and if it leads to sustained behaviour change. In addition, there is a need to explore what would motivate food retailers to offer plant-based meal options and what the impact it may have on the food purchasing behaviour.

The main strength of this review is that a systematic approach was used and reported following the PRISMA guidelines to synthesise the evidence on the interventions aiming to promote protein intake from low ecological footprint sources. We included individual and micro-environmental level behaviour change interventions, which provides a more comprehensive picture on the effectiveness of interventions in changing people behaviour towards increase in environmentally sustainable protein intake. One limitation of this review is that most of the included studies have been conducted in high income countries and only a few studies were conducted in low- and middle-income countries (LMIC). This might be due to the fact that plant-based diet concept in high income countries has received increased attention in the last five years and LMICs have not prioritised it as a significant nutrition and environmental issue due to dealing with other diet related issues such as undernutrition and nutrient deficiencies. Research indicates that meat intake in LMIC has been associated with wealth as the rise in income has resulted in significant animal-derived food consumption in these countries [85]. Furthermore, most studies used self-reported measures to measure dietary behaviours which may increase biases [86]. Also, this review was limited to the literature published in English language and did not included articles published in grey literature, therefore it may be we missed some important research written in other languages. Finally, the majority of individual level behaviour change interventions included people who may be highly motivated to change their dietary behaviour such as cancer survivors, people at risk of developing chronic conditions, limiting the generalizability of the data to general population.

## Conclusions

The present review identified effective individual and micro-environmental behaviour change interventions which showed promising results in reducing protein intake from high ecological footprint sources. The findings suggest that individual behaviour change interventions such as education, counselling and self-monitoring interventions might be useful strategies to educate people to change their dietary behaviours to more sustainable ones. However, there is a need to test these strategies among the general population longitudinally. In addition, our findings showed that altering food environments using nudging and choice architecture approaches can achieve positive dietary changes but there is a need for development and evaluation of interventions in general settings (macro-environments) and explore motivations in sustainable food purchasing behaviours. Our findings inform future research for development and evaluation of interventions and strategies to encourage greater adoption of sustainable and healthy diets.

## Supporting information

**S1 Checklist. Prisma checklist.**
(DOC)

## Author Contributions

**Conceptualization:** Rimante Ronto, Stephanie Godrich, Mark Lawrence, Shawn Somerset, Jessica Fanzo, Josephine Y. Chau.

**Formal analysis:** Rimante Ronto, Golsa Saberi, Gianna Maxi Leila Robbers, Stephanie Godrich, Josephine Y. Chau.

**Supervision:** Rimante Ronto, Mark Lawrence, Shawn Somerset, Jessica Fanzo, Josephine Y. Chau.

**Writing – original draft:** Rimante Ronto, Golsa Saberi, Gianna Maxi Leila Robbers, Josephine Y. Chau.

**Writing – review & editing:** Stephanie Godrich, Mark Lawrence, Shawn Somerset, Jessica Fanzo.

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
