## [Decision Letter · Decision Letter 0]

14 Dec 2021

PGPH-D-21-00991

Identifying effective interventions to promote consumption of protein-rich foods from lower ecological footprint sources: a systematic literature review

Dear Dr. Ronto,

Thank you for submitting your manuscript to PLOS Global Public Health. After careful consideration, we feel that it has merit but does not fully meet PLOS Global Public Health’s publication criteria as it currently stands. Therefore, we invite you to submit a revised version of the manuscript that addresses the points raised during the review process.

We look forward to receiving your revised manuscript.

Kind regards,

Changwoo Han, M.D., Ph.D.

Academic Editor

Journal Requirements:

1. Please amend your detailed Financial Disclosure statement. This is published with the article, therefore should be completed in full sentences and contain the exact wording you wish to be published.

i) Please include all sources of funding (financial or material support) for your study. List the grants (with grant number) or organizations (with url) that supported your study, including funding received from your institution. 

ii). State the initials, alongside each funding source, of each author to receive each grant.

iii). State what role the funders took in the study. If the funders had no role in your study, please state: “The funders had no role in study design, data collection and analysis, decision to publish, or preparation of the manuscript.”

iv). If any authors received a salary from any of your funders, please state which authors and which funders.

2. Please update the completed 'Competing Interests' statement. If you have no competing interests to declare, please state “The authors have declared that no competing interests exist”.

3. Please provide separate figure files in .tif or .eps format only and ensure that all files are under our size limit of 20MB.  

Once you've converted your files to .tif or .eps, please also make sure that your figures meet our format requirements.

4. We have noticed that you have uploaded supporting information but you have not included a list of legends.  Please add a full list of legends for all supporting information files (including figures, table and data files) after the references list.

Additional Editor Comments (if provided):

Reviewers' comments:

Reviewer's Responses to Questions

**Comments to the Author**

1. Does this manuscript meet PLOS Global Public Health’s publication criteria? Is the manuscript technically sound, and do the data support the conclusions? The manuscript must describe methodologically and ethically rigorous research with conclusions that are appropriately drawn based on the data presented.

Reviewer #1: Yes

Reviewer #2: Yes

2. Has the statistical analysis been performed appropriately and rigorously?

Reviewer #1: N/A

Reviewer #2: Yes

3. Have the authors made all data underlying the findings in their manuscript fully available (please refer to the Data Availability Statement at the start of the manuscript PDF file)?

Reviewer #1: Yes

Reviewer #2: Yes

4. Is the manuscript presented in an intelligible fashion and written in standard English?

Reviewer #1: Yes

Reviewer #2: Yes

5. Review Comments to the Author

Reviewer #1: Dear Author(s),

Thank you very much for your efforts, your study provides useful information for the communities and societies to understand more about the effectiveness of the interventions to promote sustainable protein consumption. I would like to have the following recommendations:

1. Data extraction and analysis (P. 9 to10): Include the name and scoring system of the quality assessment tool, as well as the determination of the quality of included studies;

2. Table 5 (P.31): Provide brief information related to the quality assessment tools, e.g. criteria to score the "selection bias";

3. Table 5 (P.31): Include the sources of funding for the studies included in the review (this is one of the criteria of one of the assessment tools named AMSTAR 2 to evaluate the quality of the systematic review, once other authors conducted the systematic review of systematic reviews in the future. Relevant information can be found at: https://amstar.ca/docs/AMSTAR-2.pdf; and

4. Results (P.10 to 30): Consider to analysis the effectiveness of interventions based on different countries or ethnicities, e.g. any countries have more significant results on the specific type of interventions. If possible.

Thank you.

Reviewer #2: The review objective was clearly stated and appropriate inclusion criteria were defined. The review was limited to Published English language studies, therefore pertinent research in other languages and gray literatures have been missed. This has to be accounted in detail in the limitation.

The reported search strategy appeared to have missed index vocabularies (MeSH/ Emtree terms), hence difficult to judge how efficient it was. No additional attempts other than database search were made to locate more articles.

The synthesis included all relevant studies. Analyses pre-defined in the methodology section were performed appropriately. Synthesis was made in qualitative manner, due to this reason it is inappropriate to assess the heterogeneity and conduct the sensitivity analyses. However, inference has to be made based on study setting. Probably the the review has be revised for high income countries. Including gray literatures could solve the skewed data for the high income countries.

The quality of the individual studies was not considered in the synthesis.

The relevance of identified studies to the review's research question appropriately considered.

6. PLOS authors have the option to publish the peer review history of their article (what does this mean?). If published, this will include your full peer review and any attached files.

**Do you want your identity to be public for this peer review?** For information about this choice, including consent withdrawal, please see our Privacy Policy.

Reviewer #1: **Yes: **Leona, Yuen-ling LEUNG

Reviewer #2: **Yes: **Kalkidan Hassen Abate

---

## [Decision Letter · Decision Letter 1]

23 Jan 2022

Identifying effective interventions to promote consumption of protein-rich foods from lower ecological footprint sources: a systematic literature review

PGPH-D-21-00991R1

Dear Dr Ronto,

We are pleased to inform you that your manuscript 'Identifying effective interventions to promote consumption of protein-rich foods from lower ecological footprint sources: a systematic literature review' has been provisionally accepted for publication in PLOS Global Public Health.

Best regards,

Changwoo Han, M.D., Ph.D.

Academic Editor

Reviewer Comments (if any, and for reference):

Reviewer's Responses to Questions

**Comments to the Author**

1. If the authors have adequately addressed your comments raised in a previous round of review and you feel that this manuscript is now acceptable for publication, you may indicate that here to bypass the “Comments to the Author” section, enter your conflict of interest statement in the “Confidential to Editor” section, and submit your "Accept" recommendation.

Reviewer #1: All comments have been addressed

2. Does this manuscript meet PLOS Global Public Health’s publication criteria? Is the manuscript technically sound, and do the data support the conclusions? The manuscript must describe methodologically and ethically rigorous research with conclusions that are appropriately drawn based on the data presented.

Reviewer #1: Yes

3. Has the statistical analysis been performed appropriately and rigorously?

Reviewer #1: N/A

4. Have the authors made all data underlying the findings in their manuscript fully available (please refer to the Data Availability Statement at the start of the manuscript PDF file)?

Reviewer #1: Yes

5. Is the manuscript presented in an intelligible fashion and written in standard English?

Reviewer #1: Yes

6. Review Comments to the Author

Reviewer #1: I have no further comment on the manuscript. However, I have one information want to clarify which is related to the assessment tool named AMSTAR 2. This is a tool also can be applied to "Review the quality of the included systematic reviews (SRs)", not on the original studies only. My suggestion to include the sources of funding for the original studies is to enhance the quality of the authors' current SR, not to recommend the authors adopted the AMSTAR 2 in the current SR (i.e. this is the "SR" not "systematic review of systematic reviews [SR of SRs]"). In addition, the attached link is for your reference to understand that once the other scholars conducting the "SR of SRs", one of the criteria to evaluate the quality of SRs included the "source of funding of the original studies included in the SRs".

7. PLOS authors have the option to publish the peer review history of their article (what does this mean?). If published, this will include your full peer review and any attached files.

**Do you want your identity to be public for this peer review?** For information about this choice, including consent withdrawal, please see our Privacy Policy.

Reviewer #1: **Yes: **Leona Yuen-ling LEUNG
